# Adversarial Feature Augmentation and Normalization for Visual Recognition

**Tianlong Chen**                                    *tianlong.chen@utexas.edu*
*University of Texas at Austin*

**Yu Cheng**                                         *yu.cheng@microsoft.com*
*Microsoft Research*

**Zhe Gan**                                          *zhe.gan@microsoft.com*
*Microsoft Cloud & AI*

**Jianfeng Wang**                                    *jianfw@microsoft.com*
*Microsoft Cloud & AI*

**Lijuan Wang**                                      *lijuanw@microsoft.com*
*Microsoft Cloud & AI*

**Jingjing Liu**                                     *JJLiu@air.tsinghua.edu.cn*
*Tsinghua University*

**Zhangyang Wang**                                   *atlaswang@utexas.edu*
*University of Texas at Austin*

**Reviewed on OpenReview:** *https://openreview.net/forum?id=2VEUIq9Yff*

## Abstract

Recent advances in computer vision take advantage of adversarial data augmentation to improve the generalization of classification models. Here, we present an effective and efficient alternative that advocates adversarial augmentation on intermediate feature embeddings, instead of relying on computationally-expensive pixel-level perturbations. We propose **A**dversarial **F**eature **A**ugmentation and **N**ormalization (A-FAN), which (*i*) first augments visual recognition models with adversarial features that integrate flexible scales of perturbation strengths, (*ii*) then extracts adversarial feature statistics from batch normalization, and re-injects them into clean features through feature normalization. We validate the proposed approach across diverse visual recognition tasks with representative backbone networks, including ResNets and EfficientNets for classification, Faster-RCNN for detection, and Deeplab V3+ for segmentation. Extensive experiments show that A-FAN yields consistent generalization improvement over strong baselines across various datasets for classification, detection and segmentation tasks, such as CIFAR-10, CIFAR-100, ImageNet, Pascal VOC2007, Pascal VOC2012, COCO2017, and Cityspaces. Comprehensive ablation studies and detailed analyses also demonstrate that adding perturbations to specific modules and layers of classification/detection/segmentation backbones yields optimal performance. Codes and pre-trained models are available in: https://github.com/VITA-Group/CV_A-FAN.

## 1 Introduction

Adversarial vulnerability is a critical issue in the practical application of neural networks. Various attacks have been proposed to challenge visual recognition models of classification, detection and segmentation (Szegedy et al., 2013; Goodfellow et al., 2014; Li et al., 2018c;b; Lu et al., 2017; Liu et al., 2018a; Lu et al., 2017;

Xie et al., 2017; Wei et al., 2018; Arnab et al., 2018; Shen et al., 2019). Such susceptibility has motivated abundant studies on adversarial defense mechanisms for training robust neural networks (Schmidt et al., 2018; Sun et al., 2019; Nakkiran, 2019; Stutz et al., 2019; Raghunathan et al., 2019; Hu et al., 2019; Chen et al., 2020; 2021; Jiang et al., 2020), among which *adversarial training* based methods (Madry et al., 2017; Zhang et al., 2019a), leveraging augmented adversarial examples, have achieved consistently superior robustness than others. However, crafting high-quality adversarial examples is computationally costly, and such adversarial training often results in a negative impact on performance over clean data (Zhang et al., 2019a).

Interestingly, a few advanced studies turn to investigate the possibility of closing networks' generalization gap (Kawaguchi et al., 2017) via adversarial training. Recent progress shows that using adversarial perturbations to augment input data/embedding can effectively alleviate overfitting issues and lead to better generalization in multiple domains, including image classification (Xie et al., 2020b), language understanding (Wang et al., 2019; Zhu et al., 2020), and vision-language modeling (Gan et al., 2020). However, it still suffers from expensive computational cost due to the generation of pixel-level perturbations when applied to image classification. We raise the following natural, yet largely open questions:

**Q1:** *Can adversarial training, as data augmentation, broadly boost the performance of various visual recognition tasks on clean data, not only image classification, but also object detection, semantic segmentation or so?*

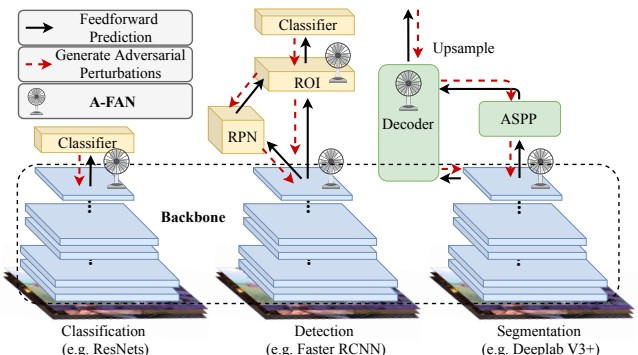

Figure 1: Overview of adversarial feature augmentation and normalization (A-FAN) for enhanced image classification (*left*), object detection (*center*) and semantic segmentation (*right*). We take ResNets (He et al., 2016), Faster RCNN (Ren et al., 2016), and DeepLab V3+ (Chen et al., 2018) pipelines as examples. Our proposed A-FAN mechanisms are plugged into backbone networks and/or ROI/decoder modules for classification/detection/segmentation, respectively.

**Q2:** *If the above answer is yes, can we have more efficient and effective options for adversarial data augmentation, e.g., avoiding the high cost of finding input-level adversarial perturbations?*

In this paper, we propose A-FAN (**A**dversarial **F**eature **A**ugmentation and **N**ormalization), a novel algorithm to improve the generalization for visual recognition models. Our method perturbs the representations of intermediate feature space for both task-specific modules (e.g., Classifiers for ResNets, ROI for Faster RCNN, and Decoder for Deeplab V3+) and generic backbones, as shown in Figure 1. Specifically, A-FAN generates adversarial feature perturbations efficiently by one-step projected gradient descent, and fastly computes adversarial features with other perturbation strengths from weak to strong via interpolation. This strength-spectrum coverage allows models to consider a wide range of attack strengths simultaneously, to fully unleash the power of implicit regularization of adversarial features.

Furthermore, A-FAN normalizes adversarial augmented features in a "Mixup" fashion. Unlike previous work (Zhang et al., 2017; Li et al., 2020) that fuses inputs or features from different samples, we amalgamate adversarial and clean features by injecting adversarial statistics extracted from batch normalization into clean features. Such re-normalized features serve as an implicit label-preserving data augmentation, which smooths the learned decision surface (Li et al., 2020). Our main contributions are summarized as follows:

- We introduce a new adversarial feature augmentation approach to enhancing the generalization of image classification, object detection, and semantic segmentation models, by incorporating scaled perturbation strength from weak to strong simultaneously.

- We also propose a new feature normalization method, which extracts the statistics from adversarial perturbed features and re-injects them into the original clean features. It can be regarded as implicit label-preserving data augmentation that smooths the learned decision boundary (illustrated in Figure 3 later on).

- We conduct comprehensive experiments to verify the effectiveness of our proposed approach over diverse tasks (CIFAR-10, CIFAR-100, ImageNet for image classification; Pascal VOC2007 and COCO2017 for object detection; Pascal VOC2007, Pascal VOC2012 and Cityspaces for semantic segmentation). The substantial and consistent performance lift demonstrates the superiority of our A-FAN framework.

## 2  Related Work

**Adversarial Attacks and Defenses.**  When presented with adversarial samples, which are maliciously designed by imperceptible perturbations (Goodfellow et al., 2014; Kurakin et al., 2016; Madry et al., 2017), deep neural networks often suffer from severe performance deterioration, e.g., Szegedy et al. (2013); Goodfellow et al. (2014); Carlini & Wagner (2017); Croce & Hein (2020) for classification models and Li et al. (2018c;b); Lu et al. (2017); Liu et al. (2018a); Xie et al. (2017); Wei et al. (2018); Zhang et al. (2020); Arnab et al. (2018); Shen et al. (2019) for detection/segmentation models. To address this notorious vulnerability, numerous defense mechanisms (Zhang et al., 2019a; Schmidt et al., 2018; Sun et al., 2019; Nakkiran, 2019; Stutz et al., 2019; Raghunathan et al., 2019) have been proposed, such as input transformation (Xu et al., 2017; Liao et al., 2018; Guo et al., 2017; Dziugaite et al., 2016), randomization (Liu et al., 2018c;b; Dhillon et al., 2018), and certified defense approaches (Cohen et al., 2019; Raghunathan et al., 2018). Among these, adversarial-training-based methods show superior robustness in defending state-of-the-art adversarial attacks (Goodfellow et al., 2014; Kurakin et al., 2016; Madry et al., 2017). Although adversarial training substantially enhances model robustness, it usually comes at the price of compromising the standard accuracy (Tsipras et al., 2019), which has been demonstrated both empirically and theoretically (Zhang et al., 2019a; Schmidt et al., 2018; Sun et al., 2019; Nakkiran, 2019; Stutz et al., 2019; Raghunathan et al., 2019).

**Adversarial Training Ameliorates Generalization.**  It is unexpected, but reasonable that recent works (Xie et al., 2020b; Zhu et al., 2020; Wang et al., 2019; Gan et al., 2020; Wei & Ma, 2019) present an opposite perspective: *adversarial training can be leveraged to enhance models' generalization if harnessed in the right manner.* For example, Xie et al. (2020b) shows that image classification performance on the clean dataset can be improved by using adversarial samples with pixel-level perturbation generation. Zhu et al. (2020) and Wang et al. (2019) apply adversarial training to natural language understanding and language modeling, both successfully achieving better standard accuracy. Gan et al. (2020) achieves similar success on various vision-and-language tasks. Parallel studies (Wei & Ma, 2019; Ishii & Sato, 2019) employ handcrafted or auto-generated perturbed features to ameliorate generalization. However, adversarial training in latent feature space as a more efficient and effective alternative has, to our best knowledge, not been studied in depth, even for classification tasks. Our work comprehensively explores this possibility not only for image classification, but also for object detection and semantic segmentation which are more challenging prediction tasks and usually require a much more sophisticated model structure, posing obstacles to easily exploit adversarial information for enhanced generalization.

**Feature Augmentation and Normalization.**  Pixel-level data augmentation techniques have been widely adopted in visual recognition models,e.g., Simard et al. (1993); Schölkopf et al. (1996); Cubuk et al. (2018); Hendrycks et al. (2019) for classification, Girshick et al. (2018); Liu et al. (2016); Zoph et al. (2019) for detection and segmentation. They are generic pipelines for augmenting training data with image-level information. Adversarial samples can also serve as a data augmentation method (Xie et al., 2020b). However, feature space augmentations have not received the same level of attention. A few pioneering works propose generative-based feature augmentation approaches for domain adaptation (Volpi et al., 2018), imbalanced classification (Zhang et al., 2019b), and few-shot learning (Chen et al., 2019).

Feature normalization plays an important role in neural network training (Ioffe & Szegedy, 2015; Li et al., 2020; Montavon et al., 2012; Li & Zhang, 1998). Ioffe & Szegedy (2015) proposes batch normalization to remove biases in the dataset, which can substantially shrink model's generalization gaps. Xie et al. (2020b) utilizes dual batch normalization to calculate statistics of adversarial and clean samples separately, therefore obtaining promising standard accuracy. Recent investigations (Ba et al., 2016; Ulyanov et al., 2016; Wu & He, 2018; Li et al., 2019; 2020) devote a particular attention to normalizing features of each training instance

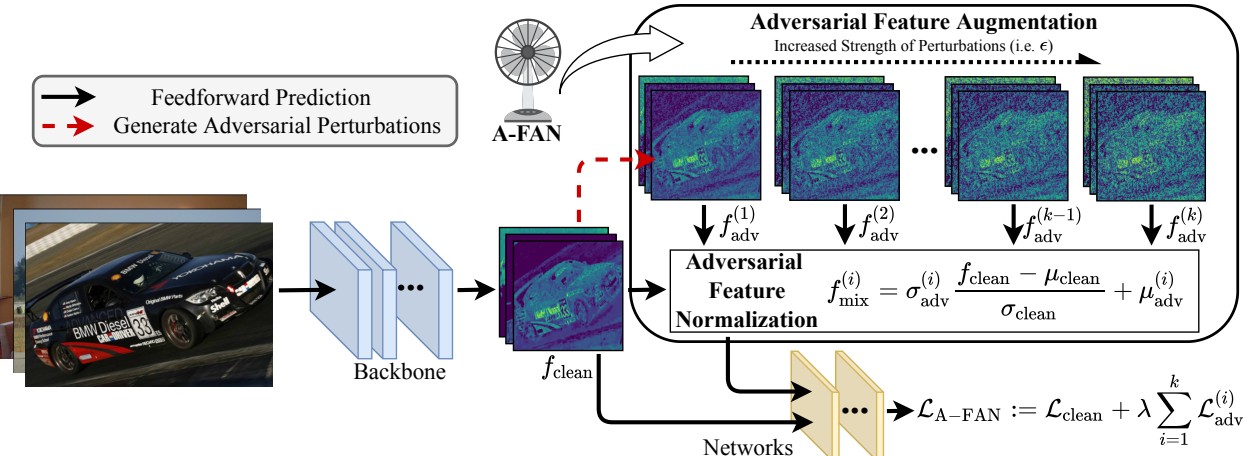

Figure 2: The pipeline of ⊛ A-FAN, which contains adversarial feature augmentation and adversarial feature normalization. From top to bottom, a series of adversarial feature perturbations with different strengths are generated to augment the intermediate clean features. Then, the statistics (i.e., $\mu_{\text{adv}}$ and $\sigma_{\text{adv}}$) of perturbed features $f_{\text{adv}}$ are extracted and re-injected into the original clean features $f_{\text{clean}}$. In the end, the normalized features $f_{\text{mix}}$ are taken as inputs by the rest of the network, and optimized by $\mathcal{L}_{\text{A-FAN}}$ with standard ($\mathcal{L}_{\text{clean}}$) and adversarial ($\mathcal{L}_{\text{adv}}$) training objectives.

individually. As an illustration, Li et al. (2020) leverages the first and second-order moments of extracted features and re-injects these moments into features from another instance by feature normalization. Different from them, we propose to utilize feature normalization techniques to combine adversarial and clean features to smooth the learned decision surface and improve model generalization.

## 3 Methodology

### 3.1 Preliminaries: Rationale of A-FAN

**Theoretical Insights.** For linear classifiers, a large output margin, the gap between predictions on the true label and the next most confident label, implies good generalization (Bartlett & Mendelson, 2002; Koltchinskii et al., 2002; Hofmann et al., 2008; Kakade et al., 2008). Although this relationship is less clear for non-linear deep neural networks, Wei & Ma (2019) establishes a similar generalization bound associated with the "all-layer margin" which depends on Jacobian and intermediate layer norms. Furthermore, Wei & Ma (2019) derives theoretical analyses that appropriately injecting perturbations to intermediate features encourages a large layer margin and leads to improved generalization. A parallel study (Wang et al., 2019) presents theoretical intuitions from a new perspective that introducing adversarial noises encourages the diversity of the embedding vectors, mitigates overfitting, and improves generalization for neural language models. These observations make the main cornerstone for our A-FAN approach valid.

**Empirical Evidences.** There exist advanced studies (Xie et al., 2020b; Zhu et al., 2020; Wang et al., 2019; Gan et al., 2020; Wei & Ma, 2019) revealing that appropriately utilizing adversarial perturbations boosts generalization of deep neural networks on diverse applications. Note that these designed approaches are *not* defense mechanisms for adversarial robustness; instead, they serve as a special data augmentation for improved performance on clean samples. Different from input perturbations (Xie et al., 2020b), our work leverages adversarial perturbation in latent feature space. To further unleash the power of adversarial augmented features, we asymmetrically fuse them with clean features, which allows the model to capture and smooth out different directions of the decision boundary (Li et al., 2020). Accordingly, A-FAN-augmented models obtain flatter loss landscape (i.e., smaller norms of Hessian with respect to model weights) and improved generalization, as supported in Table 1 and Figure 3.

Table 1: Performance and Hessian properties of ResNet-56s with or without A-FAN on CIFAR-10. A smaller spectral norm or trace of Hessian indicates a flatter loss landscape w.r.t. model weights.

| Settings | ResNet-56s | ResNet-56s + A-FAN |
|---|---|---|
| Standard Accuracy | $93.58 \pm 0.16$ | $94.83 \pm 0.11$ |
| Spectral Norm of Hessian | $23.36 \pm 1.63$ | $12.56 \pm 2.04$ |
| Trace of Hessian | $245.24 \pm 10.87$ | $208.94 \pm 7.40$ |

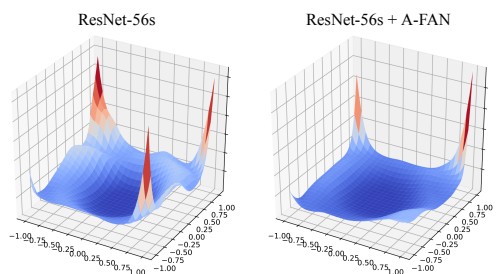

Figure 3: Loss landscape (Li et al., 2018a) of ResNet-56s with or without A-FAN on CIFAR-10.

## 3.2 Notations

Our proposed A-FAN framework includes two key components: (*i*) adversarial feature augmentation; and (*ii*) adversarial feature normalization, as shown in Figure 2. Note that we introduce adversarial perturbations in the intermediate feature space, instead of manipulating raw image pixels as in common practice.

Let $\mathcal{D} = \{\mathbf{x}, \boldsymbol{y}\}$ denotes the dataset, where $\mathbf{x}$ is the input image and $\boldsymbol{y}$ is the corresponding ground-truth (e.g., one-hot classification labels, bounding boxes or segmentation maps). Let $f(\mathbf{x}; \boldsymbol{\theta})$ with $\boldsymbol{\theta} = (\boldsymbol{\theta}_b, \boldsymbol{\theta}_t)$ represent the predictions of neural networks, where $\boldsymbol{\theta}_b$ and $\boldsymbol{\theta}_t$ are the parameters of the backbone networks and task-specific modules, respectively. For example, $\boldsymbol{\theta}_t$ denotes the parameters of ResNets' classifiers; or the parameters of RPN, ROI, and classifier in Faster RCNN; or the parameters of ASPP and Decoder in Deeplab V3+. Adversarial training (Madry et al., 2017) can be formulated as follows:

$$\min_{\boldsymbol{\theta}} \mathbb{E}_{(\mathbf{x}, \boldsymbol{y}) \in \mathcal{D}} \left[ \max_{\|\boldsymbol{\delta}\|_{\mathrm{p}} \leq \epsilon} \mathcal{L}_{\mathrm{adv}}(f(\mathbf{x} + \boldsymbol{\delta}, \boldsymbol{\theta}); \boldsymbol{y}) \right], \tag{1}$$

where $\boldsymbol{\delta}$ is the crafted adversarial perturbation constrained within a $\ell_{\mathrm{p}}$ norm ball centered at $\mathbf{x}$ with a radius $\epsilon$. The radius $\epsilon$ is the maximum magnitude of generated adversarial perturbations, which roughly indicates the strength of perturbations (Madry et al., 2017). $\mathbb{E}_{(\mathbf{x}, \boldsymbol{y}) \in \mathcal{D}}$ takes the expectation over the empirical objective over the dataset $\mathcal{D}$. The perturbation $\boldsymbol{\delta}$ can be reliably created by multi-step projected gradient descent (PGD) (Madry et al., 2017) (taking $\|\cdot\|_{\infty}$ perturbation for example):

$$\boldsymbol{\delta}_{t+1} = \Pi_{\|\boldsymbol{\delta}\|_{\infty} \leq \epsilon} \left[ \boldsymbol{\delta}_t + \alpha \cdot \mathrm{sgn}(\nabla_{\mathbf{x}} \mathcal{L}_{\mathrm{adv}}(f(\mathbf{x} + \boldsymbol{\delta}_t, \boldsymbol{\theta}); \boldsymbol{y})) \right], \tag{2}$$

where $\alpha$ is the step size of inner maximization, sgn is the sign function, and $\mathcal{L}_{\mathrm{adv}}$ is the adversarial training objective calculated over perturbed images.

## 3.3 Adversarial Feature Augmentation

In this section, we present the proposed adversarial feature augmentation mechanism. Specifically, perturbations are generated in the intermediate feature space via PGD (taking features from backbone $\boldsymbol{\theta}_b$ for example):

$$\min_{\boldsymbol{\theta}} \mathbb{E}_{(\mathbf{x}, \boldsymbol{y}) \in \mathcal{D}} \left[ \mathcal{L}_{\mathrm{clean}} + \lambda \max_{\|\boldsymbol{\delta}\|_{\infty} \leq \epsilon} \mathcal{L}_{\mathrm{adv}}(f(\mathbf{x}, \boldsymbol{\theta}_b) + \boldsymbol{\delta}; \boldsymbol{\theta}_t; \boldsymbol{y}) \right], \tag{3}$$

where the type of $\mathcal{L}_{\mathrm{clean}} = \mathcal{L}(f(\mathbf{x}, \boldsymbol{\theta}); \boldsymbol{y})$ and $\mathcal{L}_{\mathrm{adv}}$ are determined by tasks (e.g., detection models adopt regression and classification loss). $\lambda$ is a hyperparameter to control the influence of adversarial feature augmentation. Perturbations $\boldsymbol{\delta}$ are generated by PGD, as shown in Equation 2, but on the features $f(\mathbf{x}, \boldsymbol{\theta}_b)$ from the backbone network ($\boldsymbol{\theta}_b$) rather than on raw input images. Note that the formulation in Equation 3 only considers single perturbation strength.

To fully unleash the powerful of adversarial augmentation in the feature space, we propose an enhanced technique that utilizes a series of adversarially perturbed features with strength from weak to strong simultaneously. In particular, we integrate the adversarial training objective with respect to the feature perturbation strength $\epsilon$ on an interval instead of a single point.

**Approximation.** Unfortunately, the integral is intractable due to the lack of an explicit functional representation for deep neural networks. We provides an approximate solution by uniformly sampling $\{\epsilon^{(1)}, \cdots, \epsilon^{(k)}\} \in [0, \mathcal{E}]$ and subsequently generating augmented features $\{f_{\mathrm{adv}}^{(1)}, \cdots, f_{\mathrm{adv}}^{(k)}\}$, as shown in Figure 2. Specifically,

$$\sum_{i=1}^{k} \max_{||\boldsymbol{\delta}||_{\infty} \leq \epsilon^{(i)}} \mathcal{L}_{\mathrm{adv}}(f(\mathbf{x}, \boldsymbol{\theta}_b) + \boldsymbol{\delta}_i(\epsilon^{(i)}); \boldsymbol{\theta}_t; \boldsymbol{y}), \tag{4}$$

where $f_{\mathrm{adv}}^{(i)} = f(\mathbf{x}, \boldsymbol{\theta}_b) + \boldsymbol{\delta}_i(\epsilon^{(i)})$ is the adversarial augmented feature embedding.

### 3.4 Adversarial Feature Normalization

In this section, we introduce the proposed adversarial feature normalization. Inspired by Zhang et al. (2017); Yun et al. (2019); Li et al. (2020), we fuse clean ($f_{\mathrm{clean}}$) and adversarially ($f_{\mathrm{adv}}$) perturbed features for each training sample. Specifically, normalized features $f_{\mathrm{mix}}$ are crafted by normalizing clean features with adversarial feature moments. This asymmetric composition across clean and adversarial features assists networks to smooth out decision boundaries and obtain improved generalization (Li et al., 2020).

Let $\mu_{\mathrm{clean}}$ and $\mu_{\mathrm{adv}}^{(i)}$ denote the first-order moment of clean feature and the $i$-th augmented adversarial feature. Similarly, $\sigma_{\mathrm{clean}}$ and $\sigma_{\mathrm{adv}}^{(i)}$ denote the corresponding second-order moment. Their feature statistics are calculated in the routine of batch normalization (Ioffe & Szegedy, 2015). Note that the statistics can also derive from other normalization approaches (Ba et al., 2016; Ulyanov et al., 2016; Wu & He, 2018; Li et al., 2019), such as instance-norm. The detailed formulation is defined as follows:

$$f_{\mathrm{mix}}^{(i)} := \sigma_{\mathrm{adv}}^{(i)} \frac{f_{\mathrm{clean}} - \mu_{\mathrm{clean}}}{\sigma_{\mathrm{clean}}} + \mu_{\mathrm{adv}}^{(i)}, \tag{5}$$

where $i \in \{1, 2, \cdots, k\}$ refers to the index of strength levels and $k$ is the number of augmented features. Normalized features $f_{\mathrm{mix}}^{(i)}$ are fed to the networks and computed as the adversarial training objective $\mathcal{L}_{\mathrm{adv}}^{(i)}$.

---

**Algorithm 1** Adversarial Feature Augmentation and Normalization (A-FAN).

---

1: **Initialize:** $f(\mathbf{x}, \boldsymbol{\theta})$ is the visual recognition model, and $\boldsymbol{\theta} = (\boldsymbol{\theta}_1, \boldsymbol{\theta}_2)$. $f(\mathbf{x}, \boldsymbol{\theta}_1)$ are intermediate features.
2: # Generate adversarial augmented features
3: Uniformly sample $k$ different perturbation strength $\{\epsilon^{(1)}, \cdots, \epsilon^{(k)}\}$ from $[0, \mathcal{E}]$.
4: Generate adversarial perturbations $\delta_1(\epsilon^{(1)})$ with PGD, according to Equation 2 and 3.
5: Apply $\delta_1(\epsilon^{(1)})$ to the intermediate features and obtain adversarial features $f_{\mathrm{adv}}^{(1)}$.
6: **for** $\epsilon^{(i)} \in \{\epsilon^{(2)}, \cdots, \epsilon^{(k)}\}$ **do**
7:     Generate other augmented features $f_{\mathrm{adv}}^{(2)}, \cdots, f_{\mathrm{adv}}^{(k)}$ via the efficient implementation in Section 3.3.
8: **end for**
9: # Generate adversarial normalized features
10: Calculate the feature statistics $\mu_{\mathrm{clean}}$, $\sigma_{\mathrm{clean}}$ and $\{\mu_{\mathrm{adv}}^{(i)}, \sigma_{\mathrm{adv}}^{(i)}\}_{i=1}^{k}$ with batch normalization (Ioffe & Szegedy, 2015).
11: **for** $i \in \{1, 2, \cdots, k\}$ **do**
12:     Inject adversarial feature statistics $\mu_{\mathrm{adv}}^{(i)}, \sigma_{\mathrm{adv}}^{(i)}$ into clean features $f_{\mathrm{clean}}$ via the normalization, and obtain normalized features $f_{\mathrm{mix}}^{(i)}$, according to Equation 5.
13: **end for**
14: Feed normalized features to the model and compute the complete objective of A-FAN in Equation 6.
15: **Return** Training objective $\mathcal{L}_{\mathrm{A-FAN}}$

---

### 3.5 Overall Framework of A-FAN

As presented in Figure 2, we first generate a sequence of adversarial perturbations with diverse strengths to augment the intermediate features. Then, we inject perturbed feature statistics into clean features by feature

normalization. In the end, the augmented and normalized features $f_{\text{mix}}^{(i)}$ together with clean features $f_{\text{clean}}$ are both utilized in the network training. In this way, adversarial training can be formulated as an effective regularization to improve the generalization of visual recognition models. The full algorithm is summarized in Algorithm 1.

After incorporating adversarial feature augmentation and normalization, the complete training objective of A-FAN can be computed as follows:

$$f_{\text{A-FAN}} := \mathcal{L}_{\text{clean}} + \lambda \sum_{i=1}^{k} \mathcal{L}_{\text{adv}}^{(i)}, \tag{6}$$

where $\lambda = 1$ is tuned by grid search[1].

## 4 A-FAN on Image Classification

**Datasets and Backbones.** We consider three representative datasets for image classification: CIFAR-10, CIFAR-100 (Krizhevsky et al., 2009), and ImageNet (Deng et al., 2009). In our experiments, the original training datasets are randomly split into 90% training and 10% validation. The early stopping technique is applied to find the top-performing checkpoints on the validation set. Then, the selected checkpoints are evaluated on the test set to report the performance. The hyperparameters are tuned by grid search, which are quite stable from validation to test sets based on our observations, including PGD steps, step size $\alpha$, the layers to introduce adversarial perturbations, and the number of perturbations with different strength levels. We evaluate large backbone networks (ResNet-18/50/101/152 (He et al., 2016), EfficientNet-B0 (Tan & Le, 2019)) on ImageNet, and test smaller backbones (ResNet-20s/56s) as well on CIFAR-10 and CIFAR-100.

**Training Details and Evaluation Metrics.** For network training on CIFAR-10 and CIFAR-100, we adopt an SGD optimizer with a momentum of 0.9, weight decay of $5 \times 10^{-4}$, and batch size of 128 for 200 epochs. The learning rate starts from 0.1 and decays to one-tenth at 50-th and 150-th epochs. We also perform a linear learning rate warm-up in the first 200 iterations. For ImageNet experiments, following the official setting in Pytorch repository,[2] we train deep networks for 90 epochs with a batch size of 512, and the learning rate decay at 30-th and 60-th epoch. The SGD optimizer is adopted with a momentum of 0.9 and a weight decay of $1 \times 10^{-4}$. We evaluate the generalization of a network with Standard Testing Accuracy (**SA**), which represents image classification accuracy on the original clean test dataset.

Table 2: Standard testing accuracy (SA%) of ResNet-20s/56s on CIFAR-10 and CIFAR-100. Baseline denotes the standard training without A-FAN. ↑ indicates the improvement over SA compared to the corresponding baseline in standard training.

| Settings | CIFAR-10 | | CIFAR-100 | |
|---|---|---|---|---|
| | Baseline | A-FAN | Baseline | A-FAN |
| ResNet-20s | 91.25 | 92.52 (↑ 1.27) | 66.92 | 67.89 (↑ 0.97) |
| ResNet-56s | 93.59 | 94.82 (↑ 1.23) | 71.22 | 72.36 (↑ 1.14) |

Table 3: Standard testing accuracy (SA%) of ResNet-18/50/101/152 and EfficientNet-B0 on ImageNet.

| Settings | ImageNet | |
|---|---|---|
| | Baseline | Baseline + A-FAN |
| ResNet-18 | 69.38 | 70.25 (↑ 0.87) |
| ResNet-50 | 75.21 | 76.33 (↑ 1.12) |
| ResNet-101 | 77.10 | 78.14 (↑ 1.04) |
| ResNet-152 | 78.31 | 78.69 (↑ 0.38) |
| EfficientNet-B0 | 77.04 | 77.50 (↑ 0.46) |

**CIFAR and ImageNet Results. We apply PGD-5 and PGD-1 to augment the feature embeddings in the last block with adversarial perturbations for CIFAR and ImageNet models, respectively.** A series of adversarial augmented features are crafted with three different strengths uniformly sampled from $[0,\alpha]$, where the step size $\alpha = 0.5/255$. Table 2 and Table 3 present the standard testing accuracy of diverse models on CIFAR-10, CIFAR-100 and ImageNet. Comparing the standard training (i.e., Baseline) with our proposed A-FAN, here are the main observations:

- A-FAN obtains a consistent and substantial improvement over standard accuracy, *e.g.*, 1.27% on CIFAR-10 with ResNet-20s, 1.14% on CIFAR-100 with ResNet-56s, 1.12% and 0.46% on ImageNet with ResNet-50 and EfficientNet-B0. This suggests that training with augmented and normalized

---

[1]The grid is {0.01, 0.1, 0.5, 0.6, 0.7, 0.8, 0.9, 1.0, 1.1, 1.2, 1.3, 1.4, 1.5, 5, 10}.
[2]https://github.com/pytorch/examples/tree/master/imagenet

Table 5: Details of training and evaluation. We use the standard implementations and hyperparameters in Ren et al. (2015); Chen et al. (2018). The evaluation metrics are also follow standards in Ren et al. (2015); Chen et al. (2018). Linear learning rate warm-up for 100 iterations is applied.

| Datasets | Detection | | Segmentation | | |
|---|---|---|---|---|---|
| | Pascal VOC2007 | COCO2017 | Pascal VOC2007 | Pascal VOC2012 | Cityspaces |
| Batch Size | 8 | 8 | 4 | 4 | 4 |
| Iterations | $11,250$ | $180,000$ | $10,000$ | $30,000$ | $30,000$ |
| Init. Learning Rate | 0.008 | 0.01 | 0.01 | 0.01 | 0.1 |
| Learning Rate Decay | $\times 0.1$ at $6,250, 8,750$ | $\times 0.1$ at $120,000, 160,000$ | Polynomial w. power 0.9 | | Polynomial w. power 0.9 |
| Optimizer | SGD with momentum 0.9 and weight decay $5 \times 10^{-4}$ | | SGD with momentum 0.9 and weight decay $1 \times 10^{-4}$ | | |
| Eval. Metric | mAP | AP, $AP_{50}$, $AP_{75}$ | mIOU | mIOU | mIOU |

features generated by A-FAN effectively enhances the generalization of deep networks. We hypothesize that it is because adversarial perturbed features are treated as an implicit regularization, leading to better solutions for network training.

- Shallow ResNets benefit more from A-FAN than deep ResNets (*e.g.*, 1.12% on ResNet-50 vs. 0.38% on ResNet-152). A possible reason is that the performance of standard trained deep ResNets is already saturated, leaving little room for improvement.

Furthermore, we notice that A-FAN advocates different steps of PGD to achieve superior performance on diverse datasets. More ablation analyses can be found in Section 7. Meanwhile, although the robust testing accuracy is not the focus of A-FAN, we report it for completeness in Section B.1.

**A-FAN vs. AdvProp.** We compare A-FAN with AdvProp (Xie et al., 2020b) on CIFAR-10 with ResNet-18, and on ImageNet with EfficientNet-B0 (Tan & Le, 2019), as presented in Table 4[3]. CIFAR-10 models are trained on a single GTX1080 Ti GPU. ImageNet (batch size 256) experiments are conducted on $2\times$ Quadro RTX 6000 GPUs with $24G\times2$ memory in total. Since for generating feature-level perturbations, only a *partial* back-propagation to the target intermediate layer is needed which brings computational saving. The results also confirm our intuition that proposed A-FAN as an effective and efficient alternative for pixel-level adversarial augmentations (e.g., AdvProp), achieves competitive performance with much more less computational cost (i.e., less running time).

Table 4: Running time per epoch and standard testing accuracy (SA%) comparison across Baseline, AdvProp, and A-FAN.

| Settings | ResNet-18 on CIFAR-10 | | EfficientNet-B0 on ImageNet | |
|---|---|---|---|---|
| | SA | Time | SA | Time |
| Baseline | 94.30 | 23s | 77.00 | 2628s |
| AdvProp | 94.52 (↑ 0.22) | 123s | 77.60 (↑ 0.60) | 13352s |
| A-FAN | 94.67 (↑ 0.37) | 56s | 77.50 (↑ 0.50) | 6237s |

## 5 A-FAN on Object Detection

**Datasets and Backbones.** We evaluate A-FAN on Pascal VOC2007 (Everingham et al., 2010) and COCO2017 (Lin et al., 2014) for object detection. COCO2017 is a large-scale dataset with more than ten times of data than Pascal VOC2007. Specifically, in Pascal VOC2007, we use the train and validation sets for training, and evaluate on test set; in COCO2017, we train models on the train set and evaluate on the validation set. All other implementation and hyperparameters are provided in Table 5. Our experiments choose the widely-used framework, Faster RCNN (Ren et al., 2015), for detection tasks. It is worth mentioning that the proposed A-FAN approach can be directly plugged into other detection frameworks without any change, which is left to future work. We conduct experiments with both ResNet-50 (He et al., 2016) and ResNet-101 (He et al., 2016) as backbone networks.

**Pascal VOC and COCO Results.** Results are presented in Table 6. All hyperparameters of A-FAN are tuned by grid search, including PGD steps, step size $\alpha$, the layers to introduce adversarial feature

---

[3]Note that the number of PGD iterations is the same for AdvProp and A-FAN, which is 5 for ResNet on CIFAR-10 and 1 for EfficientNet-B0 on ImageNet, since these configurations lead to the best performing AdvProp (Xie et al. (2020a)'s Table 2).

Table 6: Performance of object detection on Pascal VOC2007 and COCO2017 datasets. Faster RCNN is equipped with ResNet-50/ResNet-101 backbone networks, respectively. Robustness is evaluated on the adversarial perturbed images (Li et al., 2018c; Xie et al., 2017) via PGD-10.

| COCO2017 | ResNet-50 | | ResNet-101 | | Pascal VOC2007 | ResNet-50 | | ResNet-101 | |
|---|---|---|---|---|---|---|---|---|---|
| | Baseline | A-FAN | Baseline | A-FAN | | Baseline | A-FAN | Baseline | A-FAN |
| AP (%) | 33.20 | 33.85 | 36.21 | 37.05 | | | | | |
| $AP_{50}$ (%) | 53.92 | 54.73 | 56.90 | 57.31 | mAP (%) | 73.96 | 75.38 | 74.32 | 75.71 |
| $AP_{75}$ (%) | 35.83 | 36.54 | 39.40 | 40.22 | Robust mAP (%) | 0.86 | 2.43 | 1.71 | 3.85 |
| Robust AP (%) | 0.00 | 0.50 | 0.20 | 0.66 | | | | | |

augmentations, and the number of perturbations with different strength levels. **We find that utilizing PGD-1 to generate adversarial feature perturbations in the last layer of backbone and ROI networks of Faster RCNN, achieves the most promising performance.** We adopt $\alpha = 0.3/255$ for Pascal VOC2007 and $\alpha = 0.5/255$ for COCO2017. For both datasets, a series of adversarial augmented features are crafted with five different strengths uniformly sampled from $[0,\alpha]$. To evaluate the robustness (*i.e.*, robust AP) of detection model (Li et al., 2018c; Xie et al., 2017), PGD-10 attack with $\alpha = 0.3/255$ and $\epsilon = 2.0/255$ is applied.

Table 6 summarizes the results of the baseline (*i.e.*, standard training) and A-FAN. More results with different training settings are provided in Section B.2. Comparing standard training with our proposed A-FAN mechanism, several major observations can be drawn:

- A-FAN consistently achieves substantial performance improvement across multiple backbones on diverse datasets. Specifically, A-FAN gains 0.65%/0.84% AP with ResNet-50/ResNet-101 on COCO2007, and 1.42%/1.39% mAP with ResNet-50/ResNet-101 on Pascal VOC2007. This demonstrates that training with adversarially augmented and normalized features crafted via A-FAN significantly boosts the generalization of detection models. A possible reason is that utilizing adversarially perturbed features as an implicit regularization for training leads to better generalization.

- Detectors trained on small-scale dataset benefits more from A-FAN. For example, Faster RCNN with ResNet-50 backbone obtains an almost double mAP[4] boost (*i.e.*, 1.42% vs. 0.81%) on VOC2007 than on COCO2017. It comes as no surprise that adversarially augmented and normalized features can be regarded as data augmentation in the embedding space and therefore perform more effectively on small-scale datasets (Shorten & Khoshgoftaar, 2019). We also notice that Faster RCNN with both shallow and deep ResNets gets a similar degree of improvement.

- Besides the enhanced generalization, detectors trained with A-FAN also receive better robustness, improved by 0.46% ∼ 0.50% robust AP on COCO2017 and 1.57% ∼ 2.14% robust mAP on Pascal VOC2007. Although the improved robustness still cannot hold a candle to adversarially trained models (Dai et al., 2016; Ren et al., 2016; Lin et al., 2017), it is an extra bonus from A-FAN.

- A-FAN can achieve similar improvements, compared to other previous/identical data augmentations (e.g., (Zoph et al., 2019)).

**Comparison with Learned Data Augmentation (LDA) for Object Detection.** A recent work (Zoph et al., 2019) presents learned, specialized data augmentation policies to improve generalization performance for detection models. Although it is independent of our proposed feature-level adversarial augmentation, we still provide comparison experiments for a comprehensive investigation, as shown in Table 7. Note that, for a fair comparison, we follow

Table 7: Performance of object detection on Pascal VOC2007 datasets. Faster RCNN is equipped with ResNet-101 backbone networks. Robustness are evaluated on the adversarial perturbed images (Li et al., 2018c; Xie et al., 2017) via PGD-10 with $\alpha = 0.3/255$ and $\epsilon = 2.0/255$.

| Metrics | ResNet-101 on Pascal VOC2007 | | |
|---|---|---|---|
| | Baseline | Baseline + A-FAN | LDA |
| mAP (%) | 76.00 | 79.66 | 78.70 |
| Robust mAP (%) | 2.59 | 5.05 | - |

---

[4]$AP_{50}$ shares the same meaning as mAP in VOC datasets (Zhang et al., 2020)

Table 8: Performance of object detection on Pascal VOC2007 and COCO2017 datasets. Faster RCNN is equipped with ResNet-50/ResNet-101 backbone networks. Robustness is evaluated on adversarially perturbed images (Shen et al., 2019) via PGD-10.

| Pascal VOC2012 | ResNet-50 | | ResNet-101 | | ResNet-50 | Pascal VOC2007 | | Cityspaces | |
|---|---|---|---|---|---|---|---|---|---|
| | Baseline | A-FAN | Baseline | A-FAN | | Baseline | A-FAN | Baseline | A-FAN |
| mIOU (%) | 71.20 | 72.21 | 73.65 | 74.91 | mIOU (%) | 61.51 | 62.83 | 76.00 | 76.43 |
| Robust mIOU (%) | 10.84 | 12.07 | 9.75 | 11.01 | Robust mIOU (%) | 6.77 | 7.06 | 0.51 | 1.11 |

the exact same setting as Zoph et al. (2019)'s. In addition to the detailed parameters, we combine the training sets of Pascal VOC 2007 and Pascal VOC 2012, and test the trained models on the Pascal VOC 2007 test set (4953 images). From Table 7, we observe that both A-FAN and LDA obtain performance improvements by 3.66% mAP and 2.70% mAP, respectively. Achieved superior performance further validates the effectiveness of our proposed A-FAN.

# 6 A-FAN on Semantic Segmentation

**Datasets and Backbones.** We validate the effectiveness of A-FAN on Pascal VOC2007 (Everingham et al., 2010), Pascal VOC2012 (Everingham et al., 2015), and Cityspaces (Cordts et al., 2016) for semantic segmentation. Among these commonly used datasets, Cityspaces is a large-scale datasets with more than ten times of data than Pascal VOC2007/2012. Specifically, in Pascal VOC2007, we use the train and validation sets for training, and evaluate ontest set; in Pascal VOC2012 and Cityspaces, we train models on the train set and evaluate on the validation set. All other implementation and hyperparameters are provided in Table 5. Our experiments adopt the popular framework DeepLab V3+ (Chen et al., 2018) with ResNet-50 (He et al., 2016) and ResNet-101 (He et al., 2016) as backbone networks, for segmentation tasks. Note that A-FAN can also be directly plugged into other segmentation frameworks without any change, which is left to future work.

**Pascal VOC and Cityspaces Results.** Results are collected in Table 8. **We adopt PGD-1 to craft adversarially augmented features with three different perturbation strengths (sampled from $[0,\alpha]$) in the last layer of backbone and the decoder networks of DeepLab V3+ with $\alpha = 1.0/255, 0.4/255, 0.3/255$ for Pascal VOC2007, Pascal VOC2012 and Cityspaces, respectively.** All hyperparameters are tuned by grid search. PGD-10 with $\alpha = 1.0/255$ and $\epsilon = 8.0/255$ is employed to measure robustness (*i.e.*, Robust mIOU) of segmentation models (Shen et al., 2019).

From the results in Table 8, we observe that Deeplab V3+ gains substantial performance improvement from A-FAN, which is consistent with the observations on detection models. First, A-FAN enhances the generalization of segmentation models by 1.01%/1.26% mIOU with ResNet-50/ResNet-101 on Pascal VOC2012, 1.32% with ResNet-50 on Pascal VOC2007, and 0.43% mIOU with ResNet-50 on Cityspaces. Second, A-FAN improves Deeplab V3+ more on Pascal VOC2007/2012 than on Cityspaces (*i.e.*, 1.01% ∼ 1.32% vs. 0.43%), where the former two datasets only have one-tenth amount of data compared to Cityspaces. Third, Training with A-

Table 9: Ablation study of A-FAN on CIFAR-10 with ResNet-56s. AFA: adversarial feature augmentation; AFN: adversarial feature normalization (*i.e.*, A-FAN = AFA + AFN). The settings of AFA and AFN are the same as the ones of classification experiments (Section 4), where AFA and AFN are applied to the features from the last residual block of ResNet-56s.

| Settings | Classification |
|---|---|
| | Standard Accuracy (%) |
| Baseline | 93.59 |
| + AFA | 94.45 (↑ 0.86) |
| + AFA + AFN | 94.82 (↑ 1.23) |

FAN yields moderate robustness improvement (*i.e.*, 0.29% ∼ 1.25% robust mIOU) for segmentation models.

# 7 Ablation Study and Analyses

**Augmentation v.s. Normalization.** To verify the effects of adversarial feature augmentation (AFA) and adversarial feature normalization (AFN) in A-FAN, we incrementally evaluate each module on CIFAR-10 for image classification, Pascal VOC2007 for object detection, and Pascal VOC2012 for semantic segmentation.

As shown in Table 9 and Table 10, AFA improves the baseline by 0.86% SA/1.13% AP/0.89% mIOU for classification, detection and segmentation, respectively. The combination of the two modules, AFA and AFN, gains a further performance boost by 0.37% SA/0.29% AP/0.12% mIOU on CIFAR-10, Pascal VOC2007 and VOC2012. These results demonstrate that each proposed component contributes to improving the generalization of detection and segmentation models, and AFA plays a dominant role in boosting performance.

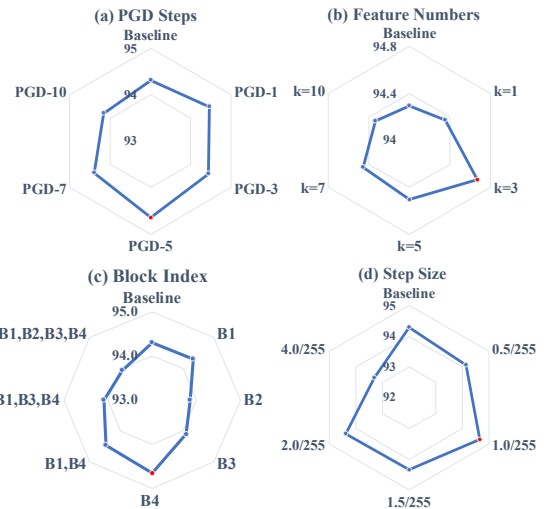

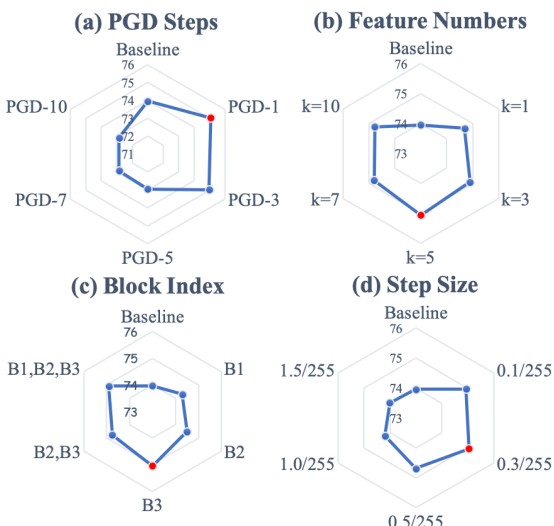

Figure 4: Ablation study on the location and strength of introducing A-FAN to classification models. Results are on CIAFR-10 dataset with ResNet-18. (a) PGD steps used in the generation of adversarial perturbations; (b) The number of augmented features, ($k$ in Equation 4); (c) The location to apply A-FAN, *e.g.*, B1 means that A-FAN is applied to features from the first residual blocks in the ResNet backbone; (d) Step size $\alpha$ that controls the strength of crafted perturbations. The red points indicate the top performance.

Figure 5: Ablation study on the location and strength of introducing A-FAN to detection models. Results are on Pascal VOC2007 dataset. (a) PGD steps used in the generation of adversarial perturbations; (b) The number of augmented features, ($k$ in Equation 4); (c) The location to apply A-FAN, *e.g.*, B1 means that A-FAN is applied to features from the first residual blocks in the ResNet backbone; (d) Step size $\alpha$ that controls the strength of crafted perturbations. The red points represent settings with top performance.

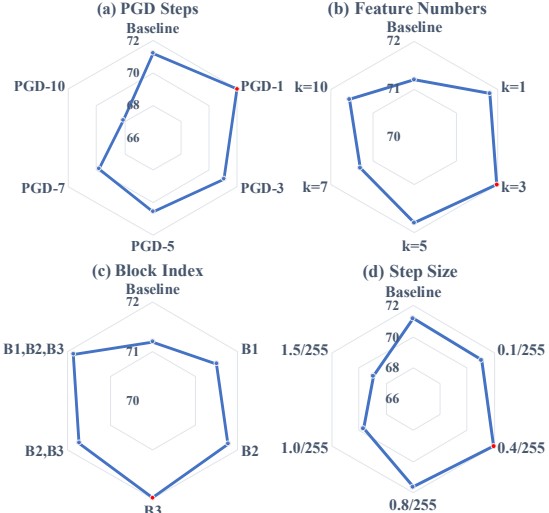

Figure 6: Ablation studies of A-FAN's location and strength on segmentation with Pascal VOC2012.

Table 10: Ablation study of A-FAN on Pascal VOC2007 and Pascal VOC2012 for detection and segmentation, respectively. AFA: adversarial feature augmentation; AFN: adversarial feature normalization (*i.e.*, A-FAN = AFA + AFN). ResNet-50 is used here and our methods are applied to the last block of backbone/ROI/Decoder. Classification is in Table 9.

| Settings | Detection | Segmentation |
|---|---|---|
|  | AP (%) | mIOU (%) |
| Baseline | 73.96 | 71.20 |
| + AFA | 75.09 (↑ 1.13) | 72.09 (↑ 0.89) |
| + AFA + AFN | 75.38 (↑ 1.42) | 72.21 (↑ 1.01) |
| A-FAN on Backbone | 75.06 (↑ 1.10) | 71.98 (↑ 0.78) |
| A-FAN on ROI/Decoder | 74.68 (↑ 0.72) | 71.71 (↑ 0.51) |
| A-FAN on Both | 75.38 (↑ 1.42) | 72.21 (↑ 1.01) |

**Effects on Backbone v.s. ROI/Decoder.** In general, detection and segmentation models can be divided into backbone and task-specific modules (*e.g.*, RPN/ROI in Faster RCNN (Ren et al., 2016) and ASPP/Decoder in Deeplab V3+). Our proposed A-FAN can be introduced to either or both modules as shown in detailed ablations in Table 10. We observe that applying A-FAN to backbone networks (↑1.10% AP/↑0.78% mIOU) gains more generalization improvement than ROI/Decoder modules (↑0.72% AP/↑0.51% mIOU) for detection and segmentation. Incorporating A-FAN on both backbone and task-specific modules always enjoys extra performance boost, compared to applying either one alone.

**Effects of Location and Strength.** The performance gain from A-FAN is determined by the location and strength of generated adversarial perturbations. Figure 4, 5 and 6 illustrate a comprehensive control study to investigate these relevant factors. Without losing generality, these ablation experiments and analyses are performed on backbone networks. When studying one of the factors, we choose the best configuration for the other factors.

To identify the proper location for A-FAN operation we inject feature perturbations to different blocks (*e.g.*, B1) or some combination of blocks (*e.g.*, B2,B3), as presented in Figure 4 (c) for classification, 5 (c) for detection, and 6 (c) for segmentation. We notice that applying A-FAN to features from the last block (*i.e.*, B3 or B4) obtains the best performance, while introducing A-FAN to multiple blocks degrades generalization.

The strength of A-FAN includes the number of PGD steps and the step size $\alpha$ for generating adversarial features, and the number of augmented features with different perturbation strengths ($k$ in Equation 4), as shown in Figure 4 (a),(b),(c) for classification, 5 (a),(b),(c) for detection, and 6 (a),(b),(c) for segmentation. Experiments show that {ResNet-18, Faster RCNN, Deeplab V3+} gains more from A-FAN with {PGD-5,PGD-1,PGD-1}, step size $\alpha = \{\frac{1.0}{255}, \frac{0.3}{255}, \frac{0.4}{255}\}$, and {3,5,3} augmented features with different perturbation strength. These systematic evaluations reveal that: (*i*) weak (*e.g.*, $\alpha = 0.1/255$) adversarial perturbed features contribute marginal generalization improvements; (*ii*) excessively strong (*e.g.*, PGD-10, $\alpha = 4.0/255$) A-FAN incurs performance deterioration. In summary, we observe that a proper configuration for A-FAN usually produces high-quality augmented and normalized features, realizing enhanced visual recognition models.

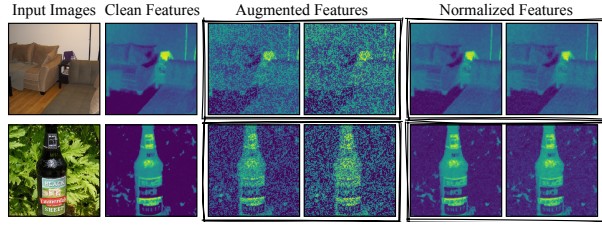

Figure 7: Visualization of adversarially augmented and normalized features for classification models with A-FAN, using a trained ResNet-18. The fifth and sixth columns are normalized features of the third and forth columns, respectively.

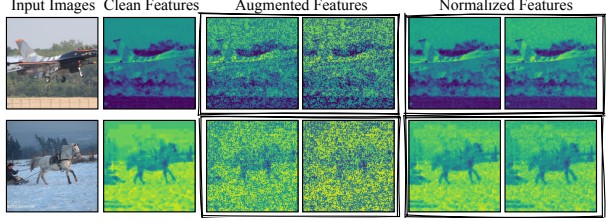

Figure 8: Visualization of adversarially augmented and normalized features for segmentation models with A-FAN, using a trained Deeplab V3+. The fifth and sixth columns are normalized features of the third and forth columns, respectively.

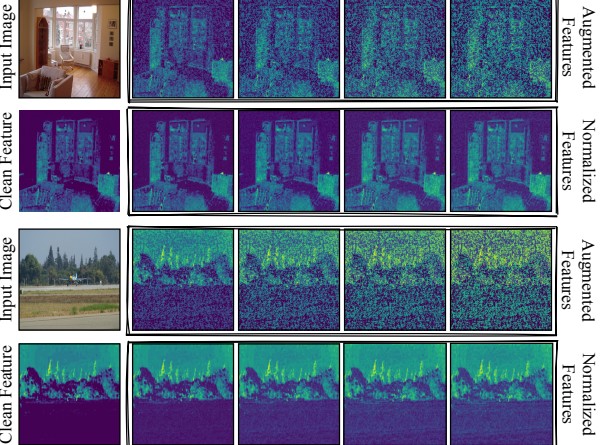

Figure 9: Visualization of adversarial augmented and normalized features for detection models with A-FAN, using a trained Faster RCNN. The left column shows the input image and the corresponding clean feature. The remaining four columns, from left to right, present features with an increased perturbation strength; from up to bottom, it shows augmented and normalized features alternatively.

**Comparing A-FAN with Random Noise.** One straight-froward approach to augment feature embeddings is injecting random noise. Here we replace the generated adversarial noise in our proposed mechanism with a randomly sampled noise from Gaussian distribution $\mathcal{N}(0, \alpha^2)$. As shown in Table 11, AFA+AFN (*i.e.*, A-FAN) achieves a larger performance gain than Random Noise+AFN, suggesting that gradient-based crafted feature augmentation benefits more to the generalization of visual recognition models.

Table 11: Performance comparison between adversarial feature perturbations with the strength $\alpha$ and random noise sampled from a Gaussian distribution $\mathcal{N}(0, \alpha^2)$. Results are reported on CIFAR-10 (with ResNet-56s), Pascal VOC2007, and Pascal VOC2012 for classification, detection, and segmentation, respectively. ↑/↓ indicates performance improvement/degradation.

| Settings | CIFAR-10 | VOC2007 | VOC2012 |
|---|---|---|---|
| | SA (%) | AP (%) | mIOU (%) |
| Random Noise + AFN | 93.36 (↓ 0.23) | 73.91 (↓ 0.05) | 71.23 (↑ 0.03) |
| AFA + AFN (i.e. A-FAN) | 94.82 (↑ 1.23) | 75.38 (↑ 1.42) | 72.21 (↑ 1.01) |

**Visualization.** Figure 7, 9 and 8 provide visualization of adversarially augmented features and normalized features generated by A-FAN. Features are collected via applying A-FAN to classification, detection and segmentation models on ImageNet, Pascal VOC2007 and VOC2012 datasets, respectively. For better visualization, we use features from the first block of backbone networks and further enlarge the magnitude of adversarial perturbations by ×20 times. We notice that normalizing features by injecting adversarial statistics into clean features, seems to neutralize the excessively generated adversarial noise. It offers an explanation for the extra performance improvement by adversarial feature normalization.

## 8 Conclusion and Discussion

In this paper, we present A-FAN, an enhanced adversarial training method to improve image classification, object detection, and semantic segmentation. By generating a series of adversarial perturbations with different strengths on feature embeddings, and fusing adversarial feature statistics with clean features, A-FAN substantially boost the recognition performance of various models across multiple representative datasets, such as CIFAR-10/100, ImageNet, Pascal VOC2007/2012, COCO2017 and Cityspaces. For future work, we would like to extend A-FAN to more tasks and provide theoretical understanding of A-FAN.

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
