# OpenReview forum: "Adversarial Feature Augmentation and Normalization for Visual Recognition"
_TMLR — Accepted by TMLR_

### Review · Reviewer_pGx2 · 2022-05-01

**Summary Of Contributions:**

This paper proposes an A-FAN method by using adversarial examples to boost the performance of various visual recognition tasks on clean examples. Specifically, adversarial features, i.e., adversarial examples generated on the intermediate feature embeddings, are employed instead of the commonly used image-level adversarial examples to accelerate the training process. Also, a simple normalization strategy is proposed.

**Broader Impact Concerns:**

There is no negative ethical implications of this work.

**Requested Changes:**

Please see the above weaknesses, where the first seven comments are recommended to be answered.

**Strengths And Weaknesses:**

*Strengths
1. The paper is well organized, which is easy to follow.
2. The experimental results are somewhat good compared with the baselines.


*Weaknesses
1. The contribution of this paper is somewhat incremental. For example, there is no significantly technical difference between the proposed AFN and MoEx [Li et al. 2020]. The authors don’t clearly illustrate the difference between them. Importantly, the motivation of using AFN and why AFN can boost the performance on clean data should be explained much clearer in an intuitive or theoretical way.
2. The setting of AdvProp in Table 4 is not clear. That is to say, the numbers of iterations for PGD are same for both AdvProp and A-FAN?
3. Some closely related works in Section of “Adversarial Training Ameliorates Generalization” should be selected as comparison methods in the experimental part.
4. The illustrations of AFA and AFN in Table 9 are not clear.  Does AFA also use two batch norms like AdvProp?
5. The results in Tables 10, 6 and 8 are doubtful and confused. For example, the results of “A-FAN on Both” are same as “Baseline + AFA + AFN” (i.e., Baseline + A-AFN). However, the authors claimed that A-AFN is performed only on the feature level to accelerate the training process. It seems that the results in Tables 6 and 8 are obtained by performing A-FAN on the entire network, which is the same as the commonly used pixel-level adversarial augmentation.
6. In fact, some works have employed a single-step FGSM to make the adversarial training more computationally efficient, such as [1][2][3], which also enjoy the similar effect as multi-step PGD. Therefore, how about using these works to achieve the goal in this work.
7. The details of random Noise are not clear. Note that for A-FAN, for each clean example, multiple different perturbations are used. How many noise examples are used for each clean example when using the random noise? In addition, it seems that Random Noise cannot improve the performance of Baseline in Table 11. This conclusion is somewhat not consistent with [4].
8. More different adversarial attacks, such as FGSM, C&W, DeepFool, are recommended to be adopted to enhance the experiments.

[1] Fast is better than free: Revisiting adversarial training. ICLR 2020.
[2] Understanding and Improving Fast Adversarial Training. NeurIPS 2020.
[3] Make Some Noise: Reliable and Efficient Single-Step Adversarial Training. arXiv 2022.
[4] Dataset Augmentation In Feature Space. ICLR Workshop 2017.

---

> ### Author Response · Authors · 2022-05-24
> **Response to Reviewer pGx2 [Cons 1-4]**
>
> Thanks for rating our paper as well-organized and our experiments as good. We point-wisely address all your concerns as below:
>
> **[Cons 1. Incremental contribution.]** We politely argue that our contributions are rich and lie in several aspects:
>
> <A new adversarial feature augmentation approach.> By incorporating adversarial features with scaled perturbation strength from weak to strong, the generalization of visual recognition is greatly enhanced.
>
> Although some pioneer studies (Wei & Ma, 2019; Ishii & Sato 2019) apply handcrafted or auto-generated perturbed features to ameliorate generalization. However, adversarial training in latent feature space as a more efficient and effective alternative has, to our best knowledge, not been studied in-depth, even for classification tasks.
>
> Our work comprehensively explores this possibility not only for image classification but also for object detection and semantic segmentation which are more challenging prediction tasks and usually require a much more sophisticated model structure, posing obstacles to exploiting adversarial information for enhanced generalization easily.
>
> <Further improving augmented features via feature normalization.> Inspired by previous normalization methods and investigations about adversarial feature distributions, we extract the statistics from adversarial perturbed features and re-inject them into the original clean features. Such fusion leads to improved augmented features and enjoys an extra performance gain.
>
> <Our proposals consistently benefit diverse visual recognition tasks, including classification, detection, and segmentation.> Comprehensive experiments have been carried out to verify the effectiveness of our proposed approach over diverse tasks (CIFAR-10, CIFAR-100, ImageNet for image classification; Pascal VOC2007 and COCO2017 for object detection; Pascal VOC2007, Pascal VOC2012, and Cityspaces for semantic segmentation). The substantial and consistent performance lift demonstrates the superiority of our A-FAN framework.
>
> **[Cons 2. Technical differences between AFN and MoEx. The motivation for using AFN and why AFN works?]** Thanks for pointing out. We will further clarify this in our revision.
>
> MoEx mixes features from two input samples, whose performance is highly dependent on the interpolation weights and exchange probability (MoEx paper’s Table 10), due to the heterogeneity among different input images. However, AFN fuses the adversarial features and clean features from the same input image, which gets rid of these sensitive factors.
>
> AFN has different motivations. From previous investigations about adversarial features [R1, R2], we see there exist significant distribution shifts between adversarial features and clean features, and naively combining them limits the performance gains. We organically fuse them by injecting statistics from adversarial features into clean features. Such normalized features unleash the power of AFA and demonstrate substantial improvements in models’ generalization.
>
> Meanwhile, AFN smoothes the learned decision surface, which produces a flatter loss landscape together with AFA, as shown in Section 3.1 (theoretical insights and empirical evidence). Improved flatness implies an enhanced generalization ability.
>
> To further convince reviewer pGx2, we conduct an empirical comparison of AFN and AFA on CIFAR-10 with ResNet-56s. We obtain Baseline : Baseline + AFA + AFN : Baseline + AFA + MoEx = 93.59 : 94.82 : 94.51, where AFN reaches a superior accuracy. A possible explanation is that AFN closes the distribution discrepancy between adversarial and clean features, obtaining extra performance boosts on top of applying AFA.
>
> [R1] Adversarial Examples Improve Image Recognition
>
> [R2] Once-for-All Adversarial Training: In-Situ Tradeoff between Robustness and Accuracy for Free
>
> **[Cons 3. The setting of AdvProp in Table 4.]** Thanks for pointing out and sorry for missing the details. Yes, the number of PGD iterations is the same for AdvProp and A-FAN, which is 5 for ResNet on CIFAR-10 and 1 for EfficientNet-B0 on ImageNet, since these configurations lead to the best performing AdvProp as presented in [R1] (Table 2).
>
> Compared to AdvProp, A-FAN has substantial computational savings, because only a partial backpropagation to the target intermediate layer is needed during the generation of feature-level perturbations. We will enrich the setup details in our revision.
>
> **[Cons 4. More comparison methods from related works.]** Great suggestion. We further implement algorithms from Wei & Ma, 2019 on CIFAR-10 with ResNet-18 and their default training configurations. We see Baseline : A-FAN : (Wei & Ma, 2019) = 94.30 : 94.67 : 94.38, where our A-FAN presents a clear advantages. More comparison experiments like Ishii & Sato, 2019 are promised in our final version.

---

> ### Author Response · Authors · 2022-05-24
> **Response to Reviewer pGx2 [Cons 5-9]**
>
> **[Cons 5. Clarify AFA and AFN in Table 9.]** NO, our AFA does not use two batch norms, while it is an interesting extension and we will explore it in the future. Besides the algorithms of AFA and AFN described in Sections 3.3, 3.4, and 3.5, we do not involve extra modules in A-FAN. The settings of AFA and AFN in Table 9 are the same as our main results on classification (Section 4), where AFA and AFN are applied to the features from the last residual block of ResNet-56s. The above clarifications will be added to our revision.
>
> **[Cons 6. Confusing results in Tables 6, 8, and 10.]** We respectfully disagree.
>
> <A-AFN is performed only on the feature of a few layers.> As described in Sections 5 and 6, “We find that utilizing PGD-1 to generate adversarial feature perturbations in the last layer of backbone and ROI networks of Faster RCNN”, and “in the last layer of backbone and the decoder networks of DeepLab V3+”. The results in Table 6 and Table 8 are obtained by performing A-FAN on the last layer of the backbone and ROI/Decoder, which is significantly different from pixel-level adversarial augmentations.
>
> <“Baseline+AFA+AFN” = “A-FAN on Both”.> As depicted in Sections 5 and 6, **we only apply A-FAN to two positions: (1) the last layer of backbones; (2) the last layer of ROI or Decoder.** “Baseline+AFA+AFN” denotes that using AFA and AFN at both locations of (1) and (2); “A-FAN on Both” also stands for leveraging A-FAN on the same two locations. This is why these two settings have the same performance. We will further clarify it in our revision.
>
> **[Cons 7. Using efficient FSGM.]** Great suggestion. It is worthy to mention that our proposed framework AFAN is compatible with most of the existing adversarial generation approaches, and can be easily plugged in. We follow reviewer pGx2’s suggestion and conduct additional experiments with the efficient adversarial training method [R3]. Under the same configurations from Table 9, we have Baseline : A-FAN w. PGD : A-FAN w. [R3] = 93.59 : 94.82 : 94.63. We see that efficient adversarial training methods like [R3] can effectively be integrated into our A-FAN, bringing more computational efficiency at a cost of slightly fewer performance gains.
>
> [R3] Understanding and Improving Fast Adversarial Training. NeurIPS 2020
>
> **[Cons 8. Details of random noises. Not consistent with [4]?]** Thanks for pointing out. We perform a fair comparison between “random noise + AFN” and “AFA + FAN” where the number of perturbed features is the same. For example, for ResNet-56s on CIFAR-10, A-FAN uses 3 augmented features, and “random noise + AFN” also adopts 3 different random feature noises for each clean example.
> We politely argue that the observations of random noise are consistent between [4] and our paper. Specifically, In [4], random noise **helps** the classification of the Arabic Digits dataset (Table 1 in [4]) but is **harmful** to the AUSLAN dataset (Table 2 in [4]). In our study, the random feature noise slightly **helps** segmentation on VOC2012, while **hurting** the classification and detection.  We see the effects of random noise can be either positive or negative based on different types of tasks, datasets, and models.
>
> [4] Dataset Augmentation In Feature Space. ICLR Workshop 2017.
>
> **[Cons 9. More different adversarial attacks.]** Thanks for the constructive suggestion. We conduct extra experiments with FSGM+GradAlign [R3] and C&W [R4]. Under the same configurations from Table 9, we have Baseline: A-FAN w. PGD: A-FAN w. [R3]: A-FAN w. [R4] = 93.59 : 94.82 : 94.63 : 93.53. We see (i) moderate adversarial perturbed features (e.g., PGD and [R3]) contribute to generalization improvements, while (ii) excessively strong A-FAN (e.g., [R4]) incurs performance degradations. More results about different adversarial attacks like DeepFool are promised in our final version.
>
> [R3] Understanding and Improving Fast Adversarial Training
>
> [R4] Towards Evaluating the Robustness of Neural Networks

---

### Review · Reviewer_aZf3 · 2022-05-26

**Summary Of Contributions:**

This paper proposes a training framework called Adversarial Feature Augmentation and Normalization (A-FAN) to improve the visual recognition models. A-FAN firstly generates a series of adversarial augmentations on intermediate feature embeddings with different levels of perturbation strength. Then, it applies the adversarial feature normalization to replace the original batch normalization statistics with the adversarial ones. The augmented new features will be fed to the network and optimized as auxiliary losses. The proposed A-FAN can boost the performance of various models across multiple representative datasets and tasks, such as CIFAR-10/100, ImageNet, Pascal VOC2007/2012, COCO2017.

**Requested Changes:**


Several concerns and suggestions for the paper:

C1) This paper claims to improve the generalization ability of models, which I, unfortunately, cannot agree with. From my perspective, the proposed A-FAN only improves the performance of models rather than the generalization ability, because it's just evaluated on the conventional metrics and I.I.D. test environments. Specifically, the generalization ability usually represents the robustness under the distribution shift, domain adaptation, transfer learning, or adversarial examples. However, these settings are not evaluated by this paper, so it's improper to claim an improvement in the generalization ability.

C2) This efficiency of the proposed method remains to be discussed. Although the proposed method only requires general adversarial samples on the feature level rather than the pixel level, it needs to generate multiple samples with different levels of the perturbation strength, which is still quite time-consuming.

S1) The title of Section 3 is supposed to be Approach rather than Preliminaries because it contains the entire proposed approach. In fact, the Preliminaries should only be the title of Sub-section 3.1.

S2) The proposed A-FAN seems to just extend the AdvProp (Adversarial Examples Improve Image Recognition, Cihang Xie, et al.) from pixel-level adversarial augmentations to feature-level adversarial augmentations. Therefore, more experimental analyses between them should be included in this paper. Yet, the performance of AdvProp is only reported in Table 4.  Is there any specific reason to explain why the performances of AdvProp are not reported in other experiments?

**Strengths And Weaknesses:**


1) This paper introduces a general feature augmentation method to improve the performance of various models across different tasks, e.g., image classification, object detection, and semantic segmentation.
2) The proposed feature normalization replaces the original feature statistics with a series of adversarial statistics with different levels of perturbation strength. Then, the method will optimize the model to ensure the same prediction across the various statistics in the normalization layer, so it may learn a more robust decision boundary against the distribution shift.
3) The comprehensive experiments on CIFAR-10, CIFAR-100, ImageNet for image classification, Pascal VOC2007 and COCO2017 for object detection, and Pascal VOC2007, Pascal VOC2012, and Cityspaces for semantic segmentation demonstrate the effectiveness of the proposed method.

---

> ### Author Response · Authors · 2022-05-26
> **Response to Reviewer aZf3 [Cons 1-4]**
>
> Thanks for acknowledging our experiments as comprehensive. We point-wisely address all your concerns as below:
>
> **[Cons 1. Performance rather than the generalization ability.]** Our generalization ability here refers to the generalization gap between training and testing performance. It follows the classic definition [R1], which assumes the unseen evaluation test-set is generated by independent and identically distributed (i.i.d.) draws according to the true distribution. Since the training performance is near perfect in most cases, we use the testing performance improvements to indicate the benefits on the generalization gap.
>
> We will replace the term “generalization ability” with “generalization gap [R1]”, and further clarify it in our revision to alleviate the confusion.
>
> [R1] Generalization in Deep Learning. Kenji Kawaguchi, Leslie Pack Kaelbling, Yoshua Bengio
>
> **[Cons 2. The efficiency of the proposed methods.]** We respectfully disagree. Our approach is more efficient than pixel-level adversarial augmentations. The evidence lie in (1) substantial running time savings; (2) an efficient implementation of A-FAN.
>
> <~50% running time savings.> In Table 4, A-FAN adopts {PGD-5, 3 augmented features} and {PGD-1, 3 augmented features} for ResNet-18 on CIFAR-10 and EfficientNet-B0 on ImageNet, respectively; AdvProp uses PGD-5 and PGD-1 to generate pixel-level augmentations accordingly. We see A-FAN achieves **54.57%** and **53.29%** running time reductions, compared to AdvProp. It is within expectation since our A-FAN is only applied to the features from the last block of the backbone or ROI or decoder, which is a superior option as suggested in Figures 4, 5, and 6.
>
> <An efficient implementation of A-FAN.> As described in Section A.1, for A-FAN with PGD-1, different augmented features can be efficiently calculated by interpolating between the clean feature and its augmented variant with the maximum perturbation. This allows A-FAN to only take PGD once rather than k times, and create other augmented features with different levels of perturbation strength at negligible extra cost.
>
> Meanwhile, from the extensive ablation studies in Figures 5 and 6, we find that A-FAN with PGD-1 works best for object detection and semantic segmentation tasks. As for classification in Figure 4, A-FAN with PGD-1 also brings performance improvements compared to baseline, and functions similarly to its PGD-5 variant.
>
> **[Cons 3. “Approach” rather than “preliminaries”.]** Thanks for the constructive suggestions. We will change the title of Section 3 to “Methodology”, and replace the title of Section 3.1 with “Preliminaries: Rationale of A-FAN”.
>
> **[Cons 4. A-FAN versus AdvProp.]** We politely argue that A-FAN is highly different from AdvProp. Specifically, (1) our feature level augmentation is domain-agnostic and more generally applicable, compared to the data level augmentation which usually requires certain domain-specific designs; (2) Compared to AdvProp, A-FAN significantly trims down the computational cost during the generation of adversarial perturbation (as shown in Table 4), since it is mainly applied to intermediate features and only needs a partial backpropagation; (3) We organically fusing augmented and clean feature by injecting statistics from adversarial features into clean features. Such normalized features unleash the power of AFA and demonstrate extra performance improvements.
>
> Meantime, we report AdvProp on the classification of CIFAR-10 and ImageNet, because the original paper only supports the classification task. Actually, we have implemented the AdvProp in detection and segmentation. However, it leads to inferior performance compared to baseline vanilla training, with default and carefully tuned hyperparameters. We will continue to investigate it and communicate with the authors in AdvProp in the future.
>
> Lastly, we conduct additional comparisons with previous feature augmentations (Wei & Ma, 2019) on CIFAR-10 with ResNet-18. We see Baseline : A-FAN : (Wei & Ma, 2019) = 94.30 : 94.67 : 94.38, where our A-FAN presents a clear advantages. More comparison experiments like Ishii & Sato, 2019 are promised in our final version.
>
> [R2] Adversarial Examples Improve Image Recognition, Cihang Xie, et al.

---

> > ### Comment · Reviewer_aZf3 · 2022-06-06
> > **Thanks for the feedback**
> >
> > I am mostly ok with the replies and appreciate the additional experiments.

---

### Review · Reviewer_tMLU · 2022-05-27

**Summary Of Contributions:**

In this submission, the author proposed a feature-level data augmentation method, realized by adversarial training. It was demonstrated to be effective on three common vision tasks including classification, detection, and segmentation. Several ablation studies were conducted and showed different impacts over the choice of location (which layer's / block's feature would be augmented) and strength.

**Requested Changes:**

* The paper used a number of text-wrapped tables, but the gap between tables and texts is too small.

* Page 4, there should be standard deviations in Table 1.

* Page 5, it is not necessary to present Eq. (4) first, as it was not used anyway. Eq. (5) is not correct as the approximated term does not have the subinterval term. Since the authors, essentially, sampled the strength terms (\epsilon) uniformly, Sec 3.3 can be simplified without the logic of Eq. (4) --> numerical integration.

* Page 6, Eq. (6) is somewhat confusing, the author may want to clarify that index "i" does not refer to feature dimension, it actually refers to the index of strengths.

* Page 6. is the (re)normalization for adversarial feature a drop-in replacement for batch normalization, if so, does it omit the rescale operation, which is usually following BN?

* Page 6. what is the grid for \lambda?

* For all experiments, it would be nice to specify the location of feature augmentation.

*  Page 7, where is "Section B.1"?

**Strengths And Weaknesses:**

* Strengths

- The paper is overall well-written, and it contains all necessary details  for understanding the design of the proposed algorithm.

- It is nice to see that three quite different vision tasks (classification, detection, segmentation) can both be benefited from the proposed method.

- It is, while being obvious, a sensible choice to lift up the sample-wise (input-wise) data augmentation to feature level, so I think the proposed method is technically correct.

* Weakness

- My main concern is whether the proposed method would save time in a well designed training system.

I would accept the claim on page 8, "A-FAN vs. AdvProp", where the author stated that "since for generating feature-level perturbations, only a partial backpropagation to the target intermediate layer is needed which brings computational saving."

However, generating input-level perturbations can be parallelized, i.e., we can use a separated worker (with periodic model sync) handling the generation of adversarial samples, so the actual training time is the same as training the same model (i.e. baseline in Table 4).

This, of course, doubles the need of computational resources. On the other hand, the proposed method can be implemented in the same way, as such, the actual training time would be the same as baseline in Table 4 (if we ignore the time cost of model sync and communication overhead).

---

> ### Author Response · Authors · 2022-05-27
> **Response to Reviewer tMLU [Cons 1-8]**
>
> Thanks for rating our paper as well-written, our experiments as comprehensive, and our techniques as correct. We point-wisely address all your concerns as below:
>
> **[Cons 1. Time savings in a well-designed training system.]** Given a well-designed training system that supports sophisticated parallelizations and ignores the time cost of model sync & communication overhead, we politely point out other advantages of A-FAN, although the training time may be similar to baselines’ under this ideal condition.
>
> <Substaintal memory savings during the multi-node parallelizations.> As also notified and acknowledged by reviewer tMLU, our A-FAN performs a partial backpropagation to the target intermediate layer (usually the last block) for the perturbation generation. Therefore, each node (besides one master node) can only store and maintain the target intermediate layer to craft the feature augmentation parallelly, while the pixel-level augmentation method will need to store and process the whole model.
>
> <Easy to parallel.> The multiple augmented features with different perturbation strengths of A-FAN are easy to be parallelized like a mini-batch training of features, during both feedforward and backpropagation stages.
>
> **[Cons 2. Text-wrapped tables.]** Thanks for pointing out. In our revision, we will reduce the number of text-wrapped tables by putting them together in a multi-column way, and enlarge the space between tables and texts.
>
> **[Cons 3. Standard deviation in Table 1.]** We follow reviewer tMLU’s suggestion and conduct five independent runs for Table 1. We obtain:
>
> |Settings|ResNet-56s|ResNet-56s+A-FAN|
>
> |Standard Accuarcy|93.58+-0.16|94.83+-0.11|
>
> |Spectral Norm of Hessian|23.36+-1.63|12.56+-2.04|
>
> |Trace of Hessian|245.24+-10.87|208.94+-7.40|
>
> With the standard deviation, our observations remain unchanged and are statistically significant.
>
>
> **[Cons 4. Equations (4), (5), and (6).]** For simplicity, we will remove the equation (4) and modify the equation (5) with only the numerical integration in our revision. Meantime, we will further clarify the index “i” and emphasize it refers to the index of strengths for the equation (6).
>
> **[Cons 5. Is (re)normalization a drop-in replacement for BN?]** It is not a drop-in replacement for BN. We apply adversarial feature (re)normalization to the clean feature and its augmented variants, which are post-BN if the front part of the network backbone contains a BN.
>
> **[Cons 6. The grid for \lambda.]** The grid is {0.01, 0.1, 0.5, 0.6, 0.7, 0.8, 0.9, 1, 1.1, 1.2, 1.3, 1.4, 1.5, 5, 10}. We will add this detail to our revision.
>
> **[Cons 7. Specify the location of feature augmentations.]** Great suggestion. Currently, we specify the location of A-FAN in Sections 4, 5, and 7 like “We apply PGD-5 and PGD-1 to augment the feature embeddings in the last block with adversarial perturbations for CIFAR and ImageNet models”, “We find that utilizing PGD-1 to generate adversarial feature perturbations in the last layer of backbone and ROI networks of Faster RCNN”, and “We adopt PGD-1 to craft adversarially augmented features with three different perturbation strengths (sampled from [0,α]) in the last layer of backbone and the decoder networks of DeepLab V3+”, respectively.
>
> We will follow reviewer tMLU’s comments and specify the location of A-FAN in each table and figure.
>
> **[Cons 8. Where is Section B.1?]** Section B.1 is provided in the appendix, as a part of our supplementary materials.

---

> > ### Comment · Reviewer_tMLU · 2022-06-08
> > **Re**
> >
> > Thanks for the detailed response and it addressed my concerns.

---

### Author Response · Authors · 2022-06-08
**General Response**

Dear AE and all reviewers,

We sincerely appreciate all reviewers’ and ACs’ time and efforts in reviewing our paper. We truly thank all for the insightful and constructive suggestions, which helped further improve our paper. We are glad that reviewers **pGx2** and **aZf3** are satisfied with our responses.

Most of the promised changes (>98%) have been included in our revised main draft and appendix, which are marked in blue. We will keep updating our draft.

We are confident that our response should have cleared the air, and we are happy to answer any additional questions and provide more information.

We really thank all reviewers’ and AEs' time and efforts again.

Best wishes,

Authors

---

### Decision · Action_Editors · 2022-07-06

**Recommendation:** Accept with minor revision

**Comment:**

The paper describes a new data augmentation technique that perturbs features (instead of inputs) adversarially by injecting various normalization statistics. The approach is general in the sense that it can be applied to several different vision tasks since the augmentation is done at the backbone output.  This is a valuable contribution that advances the state of the art for generalization in computer vision.  The reviewers unanimously recommend acceptance.  Well done! The authors are encouraged to incorporate the promised changes that have not been incorporated yet in the final version of the paper.